# Navigating Life with Posterior Urethral Valves—Sexual Health and Lower Urinary Tract Symptoms

**DOI:** 10.3390/jcm13154380

**Published:** 2024-07-26

**Authors:** Pirmin I. Zöhrer, Franziska Vauth, Anke K. Jaekel, Wolfgang H. Rösch, Aybike Hofmann

**Affiliations:** 1Department of Pediatric Urology, Clinic St. Hedwig, University Medical Center, 93049 Regensburg, Germany; 2Department of Neuro-Urology, Clinic for Urology, University Hospital Bonn, 53127 Bonn, Germany

**Keywords:** congenital lower urinary tract obstruction, incontinence, self-efficacy, kidney function, relationship

## Abstract

**Background:** Quality of life (QoL) is crucial for young adults with posterior urethral valves (PUV). This study investigates the impact of lower urinary tract symptoms (LUTS) on their quality of life and sexual health, including self-efficacy. **Methods:** Patients aged 16 and older treated for PUV completed four validated questionnaires (Sexual Self-Efficacy Scale (SSES-E), ICIQ MLUTS, ICIQ MLUTSsex, ICIQ LUTSqol) and an individual health questionnaire. **Results:** Eighteen (52.9%) patients responded, with a median age of 23 years (IQR 18–26). Three had terminal renal failure; two were transplanted. Thirteen urinated naturally; five used a stoma. Sixteen had mild and two had moderate LUTS. Fifteen patients completed the SSES-E, scoring an average of 80, similar to the healthy cohort (83). Renal failure or catheterization did not significantly affect the overall score. In the ICIQ MLUTSsex, patients reported no significant impact of LUTS on sexuality. However, those with moderate LUTS had lower self-efficacy than those with mild symptoms (mean 75 vs. 84). **Conclusions:** Although quality of life and sexual function do not appear to be significantly impaired, LUTS are common and appear to be associated with a decreased SSE in our cohort. This should be particularly considered during the transition to adult care.

## 1. Introduction

Advancements in pre- and postnatal diagnostics, as well as kidney and bladder management, have significantly improved the prognosis for boys with posterior urethral valves (PUV) over the past decades [1]. Consequently, a significantly larger number of patients are reaching adulthood. Beyond purely medical considerations, increased attention is being paid to daily life, overall quality of life (QoL), and sexual satisfaction.

Several studies, mainly conducted within Northern European healthcare systems, have examined the impact of PUV on aspects such as lower urinary tract symptoms (LUTS), QoL, fertility, and sexuality [1,2,3]. It is known that adults with PUV have increased incidences of LUTS, and that continence plays a pivotal role in facilitating participation in social activities. Epidemiological and clinical studies have demonstrated that sexual function, as well as paternity rates and sperm parameters, fall within the normal range [4]. Overall, QoL does not seem to be significantly impaired, although this largely depends on the presence of renal failure [5].

Sexuality and sexual health are critically important during adolescence, young adulthood, and later stages of life, irrespective of existing medical conditions [6]. Most studies on urethral valves evaluate sexuality and sexual health solely through standardized questionnaires on erectile dysfunction [5,7], which do not fully capture the complexity of sexual experiences.

In recent years, the concept of self-efficacy (SE) has gained increasing attention in scientific research. SE refers to an individual’s confidence in their ability to manage and overcome challenging situations. When it comes to chronic illness, SE is a crucial factor that affects a patient’s ability to manage their symptoms effectively [8]. In the context of sexuality, this encompasses one’s confidence in their sexual functioning, behavior, and cognitive and affective dimensions of sexuality [9].

Our study aims to investigate the influence of various determinants, such as continence, micturition, and renal function, on sexual satisfaction, QoL, and, particularly, sexual self-efficacy (SSE) in adolescents and young adults with PUV. This focus is crucial, as SE has been identified as a predictor of overall well-being, disease perception, and problem-solving capabilities.

## 2. Materials and Methods

Patients with a diagnosis of PUV who have been treated in our clinic in the last 20 years and were ≥16 years old were included. Exclusion criteria were concomitant genital malformations that could be expected to have an impact on sexuality or LUTS. In addition, patients with syndromes that could be expected to influence the ability to answer the questionnaire or affect LUTS were excluded. To ensure the accuracy and reliability of the data, patients who did not have native-level proficiency in the German language were excluded from the study. This was done to prevent any potential misunderstandings or misinterpretations of the questions posed to the participants. Eligible patients were contacted to participate in the study. In addition to the written survey, patients were also contacted by telephone. The participants were provided with an individualized questionnaire addressing various aspects of kidney function, hypertension, voiding, and fertility. Furthermore, four validated questionnaires were sent to the participants, including the ICIQ MLUTS, ICIQ MLUTSsex, ICIQ LUTSqol, and the Sexual Self-Efficacy Scale (SSES-E). The ICIQ MLUTS is a 13-item questionnaire that assesses LUTS. It generates total and subscale scores for voiding (up to 20 points) and incontinence (up to 24 points), along with a bother score (up to 130 points). The respondents were required to rate the level of bother associated with each item on a scale from 0 to 10. The total values were then categorized into mild, moderate, or severe symptoms. Values up to 16 were defined as mild, values between 17 and 25 as moderate, and values from 26 as severe symptoms [10]. The ICIQ MLUTSsex evaluates erectile function, ejaculation, pain, and the impact of LUTS on sexual life in a manner analogous to the ICIQ MLUTS, with total and bother scores provided [11]. The ICIQ LUTSqol assesses the impact of LUTS on QoL through 20 items, generating total scores ranging from 19 to 76 points and a bother score (up to 200 points). The SESS-E questionnaire, which was established in 1985 [12], assesses male sexual experience with 25 questions, which were validated in German [9]. Each question is scored on a scale of 10 to 100, with higher scores indicating greater severity of erectile dysfunction. The German version of the questionnaire includes subscales for erection-related SE, ego-related SE, and interpersonal relationships.

The statistical analysis was conducted using SPSS (Version 27, IBM, Armonk, NY, USA) and GraphPad Prism (Version 5, GraphPad Software). Baseline characteristics are presented as mean ± standard deviation or median (interquartile range), and as a number (percentage) for qualitative variables. Mann–Whitney U-tests were employed to assess the dependencies of questionnaire scores on nominal values. Spearman’s correlations were employed to identify correlations between ordinal scores, while one-sample *t*-tests were used to compare the results with those of other study cohorts.

## 3. Results

Of the total 34 patients who were contacted, 18 patients (53%) responded. The ICIQ was completed by all patients. Three patients did not complete the SSES-E (17%). The median age was 23 years (18–26 years), with the patients who did not complete the SSES being under 18 years old. Of the patients who did not return the questionnaires, eight patients could not be contacted due to a lack of current contact details. Six patients declined to participate in the survey due to the personal nature of the questions. Two patients were about to undergo major surgery, which would have placed an undue burden on their participation. Details of the patient selection are shown in the flow chart (Figure 1). Table 1 shows the demographic distributions of the subgroups in our patient population results from the individual questionnaire.

The median ICIQ-MLUTS total score was nine (6–14.25), with a bother score of 7.5 (1–21.75). Sixteen patients exhibited mild symptoms, while two exhibited moderate symptoms; none exhibited severe symptoms. The median voiding score was 5 (2.75–7.5), while the incontinence score was 0 (0–4). The voiding function was mostly only minimally affected, with patients reporting minimal distress from pressing during micturition (*n* = 12), delay (*n* = 12), and weakened stream (*n* = 6) in comparison to residual urine sensation (*n* = 13)—for each item, only one patient exhibited a bother score of at least six. These values were consistently observed in different patients. The prevalence of urinary incontinence (*n* = 3), as well as stress (*n* = 3) and urge incontinence (*n* = 4), was low, with post-micturition dribbling being the most frequently reported symptom (*n* = 5). The frequency of incontinence occurred only occasionally in any case. Fifty percent of patients experienced high micturition frequency (11–12 times) and nocturia (1–2 times) without perceiving them as bothersome. Detailed data are shown in Figure 2.

The median ICIQ LUTSqol was 22 (19.75–33.25), with a bother score of 12 (0.75–55). No patients required the use of pads for urinary control. Although the majority of patients denied significant restrictions on their daily lives, such as household (*n* = 2), job (*n* = 6), and sport (*n* = 5), half reported occasional feelings of depression, and five reported sleep disturbances. Even if only a few patients indicated that the items social life (*n* = 2), friendship (*n* = 4), partnership (*n* = 3), family life (*n* = 2), and sex (*n* = 4) have an influence, individual patients report a high level of stress from these without a clear connection to the severity of the symptoms reported. Figure 3 compares mean values of ICIQ-LUTSqol and respective bother scores.

The ICIQ MLUTSsex score IQR ranged from 0 to 1.25, with a bother score of 0 to 4 (median 0 for both). Three patients indicated that LUTS had only a minimal impact on their sexual lives, while the remaining patients denied any effects. While erection was not affected in 17 patients, 3 patients each reported occasional pain during ejaculation and reduced ejaculation. One patient reported anejaculation. Except for one patient, neither pain nor reduced ejaculation appeared to be bothersome. The detailed data are shown in Figure 4.

As an internal validation, all three ICIQ questionnaires were correlated with each other. This was significant in all cases. The questions were answered consistently by the participants. The significance levels and detailed data are shown in Figure 5.

The mean SSES-E score was 80.2 (range: 46–100), with notably high mean scores for questions regarding erection during masturbation (96), foreplay (91.33), and the beginning of sexual intercourse (92). The mean confidence in achieving orgasm during intercourse (62.67) or controlling ejaculation (66) was lower, and patients felt less able to refuse sexual advances (47.33) or request specific stimulation from partners (66.67). To stratify risk factors, the questionnaire results were compared according to subgroups (Table 2). To enhance comparability, the study population was divided into two groups with similar ages, setting an age limit of 24 years. However, none of the questionnaires analyzed showed a relevant difference between the age groups. Relationship status did not seem to have an influence on the ICIQ questionnaires, but the SSE of patients with relationships was better, and ego-related SSE was even significant (*p* = 0.003). Conversely, follow-up operations showed a negative influence on the ICIQ questionnaires in contrast to the SSES-E. Thus, follow-up operations in our cohort significantly influenced the LUTS-related QoL (*p* = 0.027).

Patients exhibiting moderate symptoms on the ICIQ-MLUTS scale exhibited lower SSES-E scores than those with mild symptoms (60.2 vs. 83.2, *p* = 0.076). The subscale pertaining to erection exhibited a statistically significant difference (62.7 vs. 87.6, *p* = 0.019). The subscales for ego-related SSE (59.2 vs. 81.3, *p* = 0.661) and interpersonal SSE (57.5 vs. 78.7, *p* = 0.661) showed no significant results.

Compared to the validation cohort of the German SSES-E questionnaire, SSE is not significantly worse in our cohort (80.2 vs. 83.8). In the sub-items, it is noticeable that erection-related (84.3 vs. 90.5) and ego-related (78.3 vs. 83.9) SE are worse compared to the control group, while interpersonal SE is slightly better (75.8 vs. 74.7). Looking at the individual questions, there are sometimes significantly lower scores for certain erection-related questions. For example, confidence in achieving a morning erection is significantly lower in patients with PUV. The mean results for each question are shown comparatively in (Figure 6).

## 4. Discussion

### 4.1. Lower Urinary Tract Symptoms

LUTS encompass a broad spectrum of complaints that can be quantified using various questionnaires [10]. For our study, we selected the validated ICIQ-MLUTS, as it captures all relevant LUTS as well as subjective burden. The majority of our patients exhibited only mild to moderate LUTS overall. Interestingly, all patients in the group who still urinated naturally reported at least one symptom.

Tikkinen et al. reported similar findings in their study of 68 patients with PUV and a control group of 272 individuals with a median age of 38 years, examining the prevalence of LUTS and subjective burden. Most patients in their study also had mild symptoms. Notably, mild hesitancy, weak stream, incomplete emptying, and straining occurred twice as often as in the control group [3]. As in our study, patients reported minimal burden from LUTS. This might be because these patients have been experiencing these symptoms since early childhood and, therefore, do not perceive them as new conditions in adulthood. This is further supported by a study that examined ICIQ-MLUTS scores in patients with a median age of 60 before and after urethral stricture surgery (preoperative 12.8, postoperative 5.01) [13]. In comparison, our patients had an average score of 10.2, which is closer to the preoperative value of the stricture patients. Two additional earlier studies also investigated the prevalence of LUTS in PUV patients. In one study, 37% of 19 patients had voiding problems but no incontinence [1]. In the second study, which included 29 patients, 52% had enuresis, 31% urinary incontinence, 45% frequency, and 52% dysuria [14].

These results further underscore the complexity of posterior urethral valve disease. Various factors can cause LUTS, such as incontinence due to sphincter damage during transurethral surgeries. The bladder itself can cause irritative symptoms like nocturia or urgency, either due to a small capacity and overactive detrusor or due to myogenic failure with high residual urine volumes [4,15,16,17]. Overflow incontinence and frequency, caused by high residual urine volumes, can be managed by intermittent self-catheterization (ISC). The predominantly moderate LUTS in our patients compared to other studies may be explained by the high rate of patients performing ISC.

### 4.2. Lower Urinary Tract Symptoms and QoL

A series of studies have already demonstrated that LUTS, particularly overactive bladder, can negatively impact social and sexual life [18,19]. However, there is only one study concerning PUV patients regarding QoL. This study was conducted by Jalkanen et al. in 2013, which found that the overall quality of life of PUV patients was not impaired compared to the general population [5]. Significantly lower scores were observed in patients with incontinence, particularly in the domains of sleep, eating, and sexual activity. Affected individuals reported increased sleep problems, mental pressure, and reduced functionality [5].

In our study, only a few patients reported in the ICIQ LUTSqol analysis that their LUTS affected daily activities such as visiting friends, sports, and other activities. However, it is noteworthy that a significant portion of our cohort occasionally reported depressive moods, nervousness, and negative emotions due to LUTS. This could be related to the frequently reported nocturia, although this connection could not be statistically proven in our cohort. The impact of chronic kidney disease on quality of life described by Jalkanen et al. [5] could also not be demonstrated in our cohort, possibly due to the small number of patients with renal insufficiency in our cohort and differences in the measurement instruments used for QoL assessment.

However, our study did find that our patients with more frequent surgeries exhibited significantly poorer QoL, which might be linked with a more severe form of the disease.

In general, however, it appears that patients with urogenital malformation do not consider lifelong symptoms to be as serious in terms of their quality of life. A study of patients with cloacal exstrophy demonstrated that neither urinary nor fecal incontinence had a significant impact on QoL [20]. It is also of interest to note that they often scored higher in terms of quality of life than a healthy comparison group [5,21,22].

### 4.3. Lower Urinary Tract Symptoms and Sex

Regarding sexual function, patients in our cohort reported no issues with erectile function based on the ICIQ-MLUTSsex questionnaire. These findings are consistent with the existing literature. In a study by Çetin et al., 39 adult males were assessed for renal, bladder, and sexual outcomes, revealing no erectile dysfunction in the 82% who completed the sexual function evaluation [23]. A comparative study by Taskinen et al. found that 6% of patients and 9% of age-matched controls reported difficulties achieving an erection, leading to the conclusion that the prevalence of erectile dysfunction (ED) in the study population mirrors that of the general population [24]. Only Lopez Pereira et al. reported mild to moderate ED in one-third of their patients [4]. In all the aforementioned studies, including our own, slow ejaculation was the most commonly reported impairment. Several factors may contribute to this: a persistent elongated prostatic urethra hindering semen transport to the penile urethra, urethral injuries from repeated transurethral procedures, a rigid bladder neck remaining open, or iatrogenic disruption of the bladder neck, leading to inadequate closure and subsequent retrograde ejaculation or anejaculation [4,23].

Notably, anejaculation was reported only rarely. In our study, only one patient occasionally experienced anejaculation. These findings are also reflected in the infertility and paternity rates described in the literature, which are comparable to those of the general population [1,4,24]. In our study, only one patient had children. The remaining patients, due to their young adult age, had not yet expressed a desire for children at the time of the survey. 

Thus, it can be inferred that manipulations of the urethra or bladder neck do not significantly affect the preservation of antegrade ejaculation and fertility [25]. This is consistent with alternative theories about the anatomical process of ejaculation that downplay the importance of bladder neck integrity [26]. Furthermore, the presence of uremia in the context of chronic renal insufficiency appears to exert a more pronounced impact on fertility than the anatomical alterations of the urethra [1]. This association was not observed in our cohort, likely due to the low rate of patients with renal insufficiency.

The significant correlation of all ICIQ questionnaires can be interpreted as internal validation. Due to the reference to LUTS in all questionnaires, it was to be expected that higher values in MLUTS would be accompanied by higher values in LUTSqol and MLUTSsex. It could be concluded from this that the test subjects consistently answered the questionnaires seriously.

In challenging situations, experiencing SSE is an important factor, particularly among young adults [6]. Studies have already shown that low SE leads to poorer outcomes in chronic diseases. In a recent study, Sezer et al. examined the impact of stoma SE on the sexual function and satisfaction of individuals with stomas whose average age was 53.76 years. The findings indicated that the development of social SE in this population can contribute positively to their sexual function and satisfaction with their sex life [27]. However, there are currently no data on SSE in any chronically ill patients [8]. 

Most studies on SSE have been conducted with adolescents who belong to vulnerable groups or are at increased risk for sexually transmitted diseases or sexual violence, often experiencing a sense of loss of empowerment [28,29]. Although not directly comparable, patients with PUV often experience a similar sense due to their complex medical histories [21]. Therefore, in this study, we have chosen to use the SESS to assess not only the traditional parameters of sexual quality of life but also the patients’ confidence in their ability to control sexual experiences [9].

For comparison, data from the age-matched validation cohort of the German SSES-E questionnaire are available [9]. These data indicate that overall SSE was not significantly lower. However, a closer examination of the subscales reveals that erection-related SSE was notably poorer in comparison to the control group, while interpersonal SSE was somewhat better. Analyzing the individual questions shows that certain erection-related items scored significantly lower. For instance, confidence in achieving a morning erection was significantly lower in patients with PUV. This lack of self-confidence contrasts with the unremarkable erections reported in the ICIQ-MLUTSsex. However, this questionnaire assesses general erections, while morning erections are known to be influenced by a variety of factors, including bladder fullness. Despite this, nocturia did not appear to negatively impact morning erections in our patient cohort. It was observed that the severity of LUTS negatively correlated with sexual self-confidence in our patient cohort, with a significant correlation noted in relation to erection-related issues. Regarding ego-related sexual self-confidence, the most significant influence was associated with relationship status. A causal link is evident, as individuals in a relationship can be expected to experience higher levels of sexual and personal affirmation. The existing literature shows a positive impact on sexual self-confidence for women in partnerships, although this effect has not been significant for men until now [9].

These findings highlight the complex interplay between physical symptoms, psychological factors, and relationship dynamics in influencing SSE, underscoring the need for a holistic approach in the management and support of patients with PUV.

Despite the prospective design and use of four validated questionnaires, our study has several limitations. We acknowledge that the number of patients in our study is a limiting factor in drawing general conclusions. However, considering the rarity of the disease and the sensitive nature of the topic, we believe we have achieved a satisfactory patient number, especially in comparison to previous studies. It should be taken into account that patients with fewer problems might have been more likely to respond. Accordingly, we had only a few patients with renal insufficiency or kidney transplants. However, about one-third of the patients had a catheterizable stoma, which is invasive and can influence the assessment of LUTS. There is no information on the current kidney function values, so no assessment can be made in this regard. However, such an evaluation would have also been beyond the scope of this study. To our knowledge, this is still one of the few studies that have comprehensively examined this topic in patients with PUV.

## 5. Conclusions

In conclusion, our study highlights that patients in our cohort typically experience mild to moderate LUTS, which impact their QoL predominantly through increased fatigue and depressive moods, despite minimal perceived disruption to daily activities. Importantly, the sexual function of our patients remains unaffected, and overall SSE mirrors that of the general population, albeit significantly influenced by the severity of LUTS and relationship status. These findings may, in addition to previous studies, underscore the need for tailored support and management strategies that address both physical symptoms and psychological well-being, particularly during transition into adulthood.

## Figures and Tables

**Figure 1 jcm-13-04380-f001:**
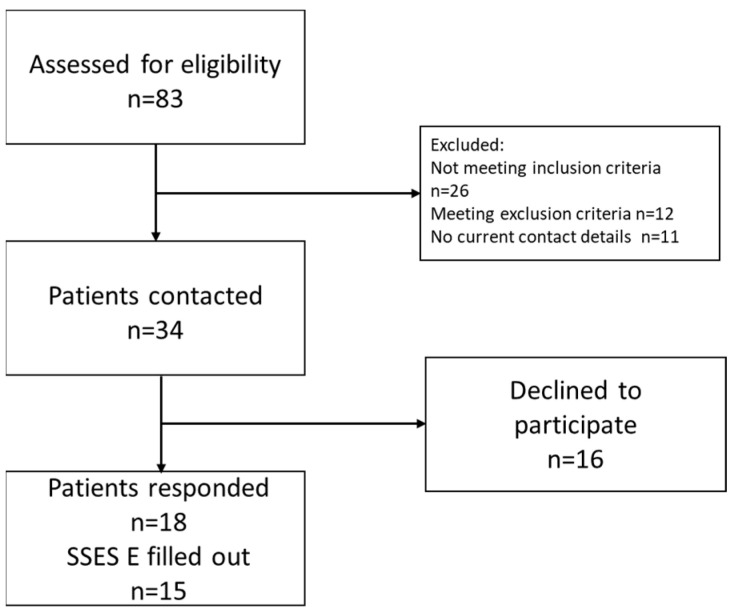
Patient selection flow chart. Of the 34 eligible and contacted patients, we received answers from 18 patients.

**Figure 2 jcm-13-04380-f002:**
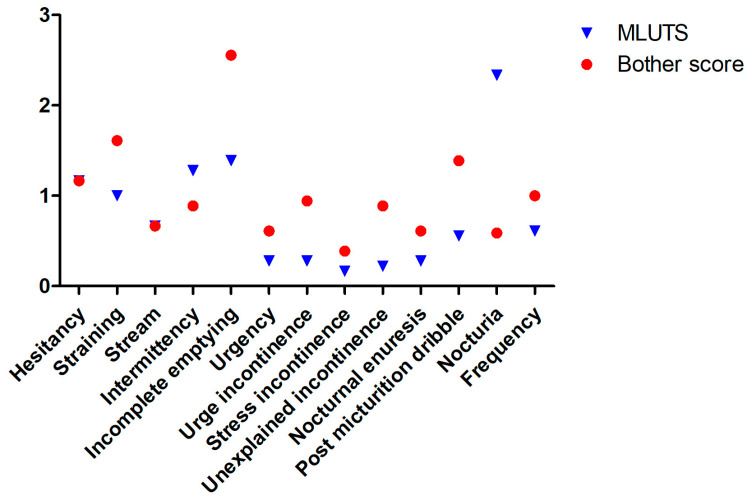
Comparison of the scores of the individual items of the ICIQ MLUTS with the respective bother scores.

**Figure 3 jcm-13-04380-f003:**
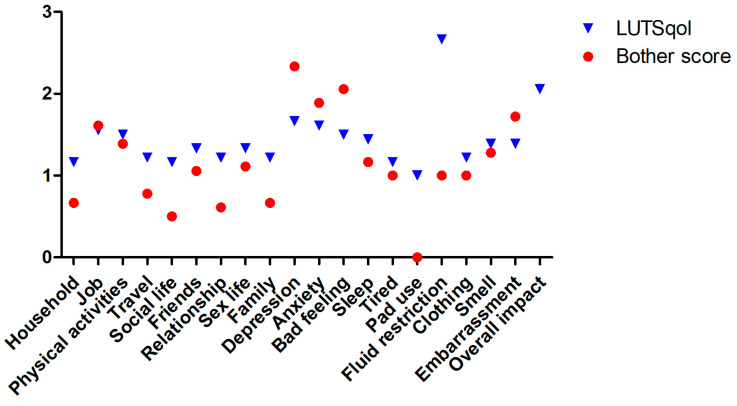
Comparison of the scores of the individual items of the ICIQ LUTSqol with the respective bother scores.

**Figure 4 jcm-13-04380-f004:**
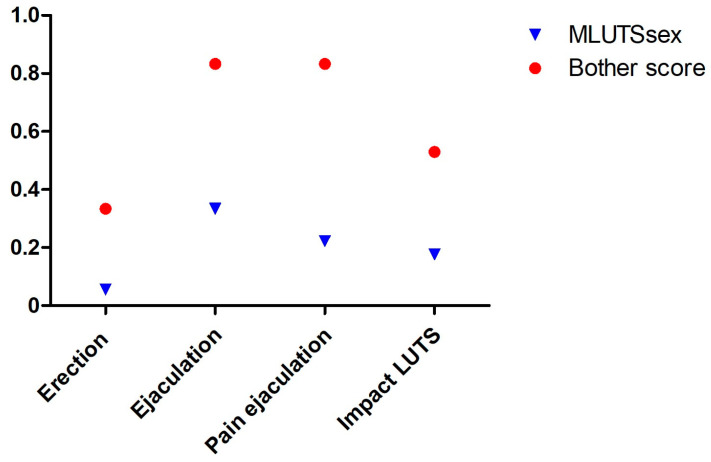
Comparison of the scores of the individual items of the ICIQ MLUTSsex with the respective bother scores. Abbreviations: LUTS (lower urinary tract symptoms).

**Figure 5 jcm-13-04380-f005:**
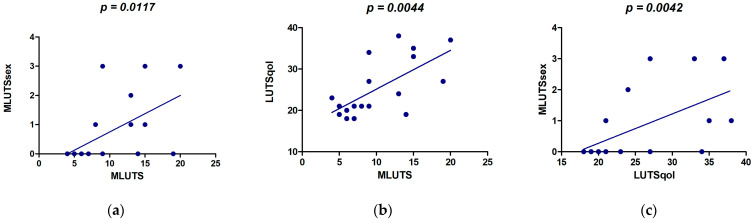
Correlation between the three ICIQ questionnaires: (**a**) correlation between ICIQ MLUTS and MLUTSsex; (**b**) correlation between ICIQ MLUTS and LUTSqol; (**c**) correlation between ICIQ LUTSqol and MLUTSsex.

**Figure 6 jcm-13-04380-f006:**
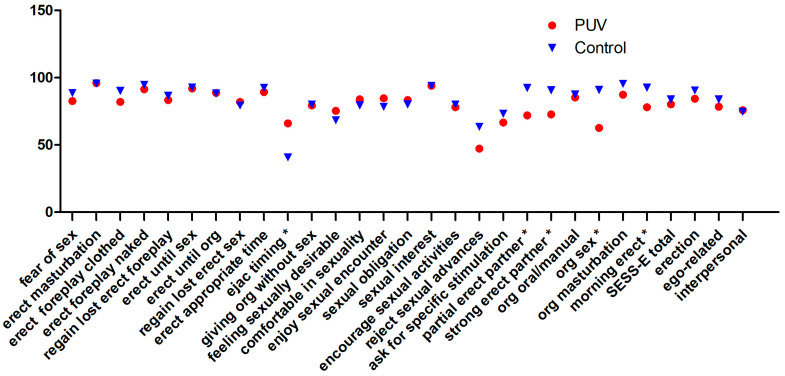
Average results of the individual SSES-E questions in our patients and from the validation cohort. Significant results (*p* < 0.05) are marked with *. Abbreviations: erect (erection), ejac (ejaculation), and org (orgasm).

**Table 1 jcm-13-04380-t001:** Demographic data. The sizes of the various subgroups covered by the individualized questionnaire are shown numerically and as a percentage. Abbreviations: IQR (interquartile range), ISC (intermittent self-catheterization), and KTx (kidney transplantation).

	*n* (%)
SSES-E filled out	15 (83)
Age median (IQR)	23 years (18–26)
Micturition via naturalis	13 (72)
ISC	5 (28)
KTx planned	1 (6)
KTx done	2 (11)
Hypertension known	6 (33)
Hypertension treated	2 (11)
In relationship	10 (56)
Follow-up operations	7 (39)
Desire to have children	0 (0)
Fatherhood	1 (6)

**Table 2 jcm-13-04380-t002:** Subgroup analysis. The detailed data, including significance levels, are listed by subgroup; significant results are marked in bold. Abbreviations: ISC (intermittent self-catheterization) and KTx (kidney transplantation).

Subgroups	ICIQ MLUTS	ICIQ MLUTSsex	ICIQ LUTSqol	SSES-E	SSES-E Erection	SSES-E Ego	SSES-E Interpersonal
Age:							
≥24y	10.7	0.6	26.3	81.65	84.2	82.71	77.34
<24y	9.625	1	24.13	78.46	84.42	73.33	74.11
*p*-value	0.829	0.515	0.237	0.694	0.694	0.281	0.694
Partnership:							
yes	10.7	0.8	25.9	85.64	87	86.33	83.25
no	9.63	0.75	24.63	69.2	78.91	62.33	61
*p*-value	0.829	0.965	0.696	0.099	0.371	**0.003**	0.055
Hypertension:							
yes	9.5	1.3	26.5	77.44	84.73	72	71,50
no	10.58	0.5	24.75	81.52	84.09	81.5	78
*p*-value	0.82	0.18	0.494	0.594	1	0.206	0.594
Operations:							
yes	10.14	1.14	30	81.76	87.1	77.33	77.75
no	10.27	0.55	22.36	79.36	82.91	78.83	74.88
*p*-value	0.86	0.328	**0.027**	1	0.768	0.768	0.953
Voiding	10	0.54	23.62	78.7	82.5	78.2	73.9
ISC	10.8	1.4	29.8	86	91.5	78.9	83.8
*p*-value	0.443	0.289	0.143	0.448	0.295	1	0.448
KTx:							
yes	10	0.67	24.33	80.2	82.73	84.17	73.75
no	11.3	1.3	30.33	80.15	84.55	77.44	76.15
*p*-value	0.654	0.426	0.36	0.933	0.686	0.476	0.476

## Data Availability

The data used to support the findings of this study are available from the corresponding author upon request.

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
