# Peer review of "Navigating Life with Posterior Urethral Valves—Sexual Health and Lower Urinary Tract Symptoms"

_jcm, 2024, doi:10.3390/jcm13154380_

Round 1
Reviewer 1 Report
Comments and Suggestions for Authors
The topic of the article – the study of health-related quality of life – is in great demand in science and practice today. In medical practice, quality of life criteria are recognized as an integral part of a comprehensive analysis of new methods of diagnosis, treatment and prevention. In today's environment, medical care is becoming more effective and life expectancy is increasing due to the development of personalized medicine programs. Therapeutic and recreational activities aimed at supporting the sustainable development of people with congenital and acquired diseases help normalize their lives. However, these people are not always satisfied with their lives, despite the improvement in their physical condition as a result of treatment. The article summarizes the results of the Young Adults with Posterior Urethral Valves (PUV) survey. This is a relatively rare disease, so having the opportunity to explore how this group feels is undoubtedly valuable. Standardized self-reporting was the primary method of data collection.
Two factors – objective (characteristics of life situation, circumstances and experiences) and subjective (needs and values, personality orientation) – determine a person's assessment of his or her quality of life. For example, respondents with the same diagnosis may rate their self-efficacy differently because of the different experiences they have had. These experiences, for instance, are relationship experiences or age-related experiences. There were respondents in the article who indicated the presence/absence of a relationship. Nevertheless, no comparative analysis between these groups is provided; instead, it is merely mentioned in passing in the Discussion section. Additionally, the conducted research included respondents between the ages of 16 and 39. However, a comprehensive analysis of the relationship between age and respondents' assessments of their quality of life was not performed. Instead, the respondents were divided into two groups: those under 24 years old and those over 24 years old, which is not entirely accurate. The study does not consider the subjective significance of different aspects of respondents' lives. Individuals with the same level of sexual functioning may assess their well-being in a particular area of life differently, depending on the significance of that area to them and their needs. Incorporating these elements into the study would undoubtedly enhance its depth.
Recommendations:
1. While the authors acknowledge the limitations of their study due to the small sample size, some of the proposed outcomes are overly categorical for the analysis of 18 cases (for instance. “Although quality of life and sexual function do not appear to be significantly impaired, LUTS are common and appear to be associated with a decrease SSE”; “However, our study did find, that patients with more frequent surgeries exhibited significantly poorer QoL, which might be linked with a more severe form of the disease”, etc.). It is essential to clarify the wording that conveys an impression of universality (attempts to extrapolate the patterns observed in 18 respondents to the entire general population).
2. In order to enhance the clarity and precision of the findings, it is recommended to document the significant relationships and differences identified (Section 5).
Reviewer 2 Report
Comments and Suggestions for Authors
The Manuscript is about posterior urethral valves and its impact on quality of life, sexual health, and LUTS. The subject is interesting but some points should be highlighted:
1) The title should be modified, avoiding abbreviations if possible;
2) in the Methods section, inclusion and exclusion criteria should be added, avoiding the total number of patients;
3) some abbreviations are not explained;
4) if baseline characteristics are presented as median, interquartile range should report instead of the total range (min - max);
5) in the Results section, a patient selection flowchart and its description should be added;
6) Table 1 should be modified, making it more clear (especially the categorical variable which should be reported as n (%));
7) Captions of all the tables and all the figures should briefly describe them, with the explanation of all the abbreviation used as well;
8) The Results section and the Discussion section should be modified in order to make them more clear and readable;
Comments on the Quality of English LanguageMinor English editing is recommended
Round 2
Reviewer 2 Report
Comments and Suggestions for Authors
The Authors followed Reviewers' suggestions and the Manuscript is more clear and suitable for publication.